# Efficiently Maintaining the Multilingual Capacity of MCLIP in Downstream Cross-Modal Retrieval Tasks

**Fengmao Lv[1], Jitong Lei[1], Guosheng Lin[2], Desheng Zheng[3,4], Jianyang Zhang[5],\* Tianrui Li[1]\***
[1]Southwest Jiaotong University, [2]Nanyang Technological University
[3]Southwest Petroleum University, [4]Kash Institute of Electronics and Information Industry,
[5]University of Electronic Science and Technology of China
`fengmaolv@126.com, 2023201818@my.swjtu.edu.cn, gslin@ntu.edu.sg,`
`zheng_de_sheng@163.com, jyzhang312@gmail.com, trli@swjtu.edu.cn`

## Abstract

While existing research on Multilingual CLIP (MCLIP) has prioritized model architecture design, our work uncovers a critical challenge in practical adaptation: fine-tuning MCLIP through a single source language risks diminishing its multilingual capabilities in downstream tasks due to cross-linguistic disparities. To bridge this gap, we systematically investigate the role of token similarity in cross-lingual transferability for image-text retrieval, establishing it as a key factor governing fine-tuning efficacy. Building on this insight, we propose two novel strategies to enhance efficiency while preserving multilinguality: 1) TaPCL dynamically optimizes training by prioritizing linguistically distant language pairs during corpus sampling, reducing redundant computation, and 2) CiPCL enriches the source corpus with multilingual key terms, enabling targeted knowledge transfer without reliance on exhaustive parallel data. By strategically balancing token similarity and domain-critical information, our methods significantly lower computational costs and mitigate over-dependence on parallel corpora. Experimental evaluations across diverse datasets validate the effectiveness and scalability of our framework, demonstrating robust multilingual retention across languages. This work provides a principled pathway for adapting MCLIP to real-world scenarios, where computational efficiency and cross-lingual robustness are paramount. Our codes are available at `https://github.com/tiggers23/TaPCL-CiPCL`.

## 1 Introduction

Cross-modal retrieval aims to retrieve the corresponding data across different modalities using a query from one modality. With the emergence of large numbers of images on social media platforms such as Twitter and Facebook, accurately retrieving multimedia contents has become a significant challenge, making research on cross-modal retrieval a hot topic. In recent years, visual-language pre-training models like CLIP [1] have gained significant attentions. Models such as CLIP effectively calibrate the visual and language modality representations during the pretraining phase, making them well-suited for tasks such as image-text cross-modal retrieval. Early research on CLIP primarily focuses on English. Considering that there are more than 7,000 languages worldwide, recent studies have extended the multilingual processing capability of the CLIP model to build Multilingual CLIP (MCLIP) models, which can support image-text retrieval applications in different languages.

However, current studies on MCLIP primarily concentrate on the construction of multilingual pre-training models, e.g., building multilingual pre-trained CLIP models through joint learning of multiple languages [2, 3, 4, 5, 6, 7, 8, 9], or expanding the multilingual capability of the CLIP model

---

\*corresponding author

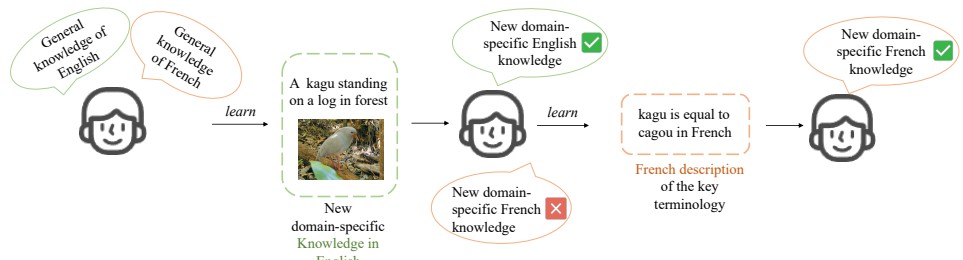

Figure 1: A person who has learned new domain-specific knowledge in English can comprehend that knowledge in French by simply learning the French description for the key terminology about the corresponding knowledge.

incrementally based on the continual learning framework [10]. In this work, we further focus on how to maintain the multilingual capability of the MCLIP model during its adaptation for downstream applications. In practical applications, MCLIP models are usually fine-tuned via one language (i.e., source language) to enhance its performance in relevant tasks. Due to inherent differences between languages, the performance improvements for other languages (i.e., target languages) are typically inferior compared to the improvement in the source language, leading to a degradation of the multilingual capability of the MCLIP model on downstream tasks. Therefore, this work will focus on multilingual generalizable fine-tuning strategies for MCLIP, enabling each language to perform well in the corresponding downstream tasks.

Similarly to existing cross-lingual learning studies [11, 12, 13], a common approach is to translate the source language corpus into target languages, and use the resulting parallel corpus of target languages alongside the source language corpus to fine-tune MCLIP models. Considering that the MCLIP model usually involves a large number of target languages, directly incorporating parallel corpora for fine-tuning will lead to a massive increase of training data for each downstream task-specific adaptation of the MCLIP model. As a result, this will cause substantial computational overhead, significantly compromising fine-tuning efficiency, and hindering the flexible deployment of MCLIP models for downstream applications. **Therefore, this work will focus on how to implement efficient fine-tuning of MCLIP models in downstream tasks, i.e., achieving fine-tuning with reduced computational time while preventing performance degradation compared to directly utilizing the parallel corpora.**

In general, visual knowledge can be transferred across similar languages. For instance, when learning a visual knowledge domain through English, a person who is proficient in multiple languages can easily understand the corresponding visual knowledge via other linguistically similar languages (e.g., French or Italian), without relearning it via French or Italian. Inspired by this observation, this work will attempt to investigate what factor can influence the knowledge transferability across languages and how to leverage the knowledge transferability across languages to reduce excessive reliance on parallel corpora, thereby lowering the computational cost of fine-tuning MCLIP models for downstream tasks, and achieving a more efficient fine-tuning.

Existing works [14, 15, 16, 17, 18, 19] primarily focus on studying cross-lingual transferability in unimodal language models such as mBERT [20] and XLM-RoBERTa [21], exploring different similarity factors (e.g., token similarity, syntactic similarity, and phonological similarity) for cross-lingual transfer in tasks such as document classification [22], natural language inference [11], and part-of-speech tagging. This work conducts the pioneer attempt to explore influential similarity factors for cross-lingual transferability in multimodal scenarios. To this end, according to some preliminary experiments on the correlation between performance improvement in target languages and different similarity factors (see Figure 2), we find that token similarity between languages significantly affects the transfer performance, i.e., target languages with higher token similarity to the source language exhibit more substantial performance improvements after fine-tuning the MCLIP model with the source language.

With the above findings, this work proposes two efficient fine-tuning strategies to enhance the computational efficiency in fine-tuning MCLIP models with parallel corpora. Firstly, we propose the Transferability-aware Parallel Corpora Learning (TaPCL) mechanism by incorporating the influence

of token similarity factors into the downstream adaptation of MCLIP models. For target languages that exhibit high token similarity to the source language, knowledge transfer from the source to the target language is more effective, so that those target languages demonstrate notable performance improvements after fine-tuning on the source language, which can be equivalent to fine-tuning on those languages. Consequently, directly utilizing all parallel corpora in target languages during fine-tuning may introduce training redundancy. Under this consideration, our approach employs language-specific sampling probabilities for different target languages during fine-tuning, rather than directly considering all target language parallel corpora. By lowering the sampling probability for languages with high token similarity to the source language, this strategy mitigates the over-utilization of parallel corpora and reduces redundant computational costs caused by redundant parallel corpora in target languages, preliminary experiments demonstrate that TaPCL improves fine-tuning efficiency.

Based on the preliminary success of TaPCL, we propose the Critical-information Parallel Corpora Learning (CiPCL) to further reduce the reliance on parallel corpora. In general, this method is inspired by the learning mechanism of humans for foreign language knowledge acquisition. Specifically, as shown in Figure 1, when a person (who is proficient in both English and French) has learned new domain-specific knowledge in English, he can also comprehend and articulate that knowledge in French by simply learning the French description for the key terminology about the corresponding knowledge, without the need to relearn the entire knowledge through French. Similarly, during pre-training, MCLIP models have already acquired syntactic structures across languages, while the primary objective of downstream task learning is to gain domain-specific knowledge. Therefore, by treating the pre-trained MCLIP model as a person who has multilingual capability, the CiPCL approach mainly focus on applying the learning principle of humans to MCLIP models. To this end, CiPCL first recognizes key information in downstream image-text retrieval tasks, and incorporates their descriptions in target languages as critical prompts in the source language corpus, e.g., "A kagu (cagou in French) standing on a log in forest". By fine-tuning MCLIP models using the source language corpus augmented with target language prompts for critical information, CiPCL does not need to specifically construct complete parallel sentences for target languages, significantly enhancing computational efficiency during fine-tuning. Similarly to our first strategy, we also employ language-specific sampling probabilities to add the target language prompts for critical information. Extensive experiments further demonstrate the effectiveness of CiPCL.

To sum up, our contributions can be summarized as follows: (1) We present an initial exploration of maintaining the multilingual capability of MCLIP models in downstream tasks. To this end, we also conduct an initial investigation to explore which similarity factors significantly impact the cross-lingual transferability in visual-language pretraining models. (2) Based on preliminary experiments and the learning mechanism of human, we propose two efficient fine-tuning strategies for MCLIP models to reduce computational time and alleviate reliance on parallel corpora while maintaining the multilingual capability of MCLIP models. (3) Extensive experiments are conducted to demonstrate both the effectiveness and scalability of our proposed methods.

## 2 Related Work

### 2.1 Multilingual CLIP Models

Models like CLIP [1] calibrate visual and language modal representations in a shared feature space, enabling cross-modal retrieval by computing the similarity between image and text representations. The initial works primarily focus on English, while subsequent works also attempt to construct CLIP models capable of handling multiple languages. Specifically, Reimers et al. [2] and Carlsson et al. [4] propose to construct multilingual CLIP by distilling knowledge from the text encoder of the original CLIP to a multilingual text encoder via parallel corpora. Chen et al. [6] propose to further distill knowledge from the vision encoder of the original CLIP to improve the image-text alignment capability of multilingual CLIP. The above works construct multilingual CLIP primarily by extending the multilingual capability of the original CLIP. OpenCLIP [5] also proposes to directly construct multilingual CLIP via pre-training from multilingual image-text data. Recently, Yang et al. [10] further propose the continual learning paradigm which enables the continuous integration of new language processing mechanisms into multilingual CLIP models. However, existing works on multilingual CLIP mainly focus on the construction of multilingual CLIP. This work will further consider how to maintain the multilingual capability of MCLIP models during its adaptation for downstream applications.

## 2.2 Cross-lingual Transfer

Fine-tuning multilingual models typically requires considering cross-lingual transfer mechanisms between different languages. Existing works primarily focus on studying cross-lingual transferability for unimodal language models. Specifically, Wu et al. [14] analyze the relationship between token similarity and cross-lingual transferability across multiple tasks, including document classification, natural language inference, named entity recognition, part-of-speech tagging, and dependency parsing. Ahuja et al. [15] develop a multitask performance prediction framework to analyze the importance of influential factors, thereby enabling more accurate analysis by leveraging data of other tasks when dealing with tasks involving limited language quantities. de Vries et al. [16] consider different multiple languages as the source languages and perform analyzes for POS tasks. Limisiewicz et al. [17] consider multiple methods and investigate the effects of tokenizer quality and token similarity on cross-lingual transferability. To further promote cross-lingual transfer when fine-tuning unimodal language models, David Schmidt et al. [18] propose to improve the performance of multilingual models on downstream tasks by integrating two independent fine-tuning phases into a unified stage, thus providing multifaceted supervision for the fine-tuning process. To maintain robust cross-lingual sentence representations with minimal alteration to the model's output representations during fine-tuning, Tu et al. [23] leverage a prompt-based structure for model adaptation. Yu et al. [24] propose leveraging mutual supervision between language models during fine-tuning to enhance cross-lingual transferability. With the recent advances in machine translation [25, 26] (e.g., language translation enhanced by LLM), cross-lingual transfer can be enhanced by directly leveraging parallel corpora translated from the source language during fine-tuning [11, 12, 13]. However, existing works mainly focus on single-modal models/tasks. For MCLIP models typically involve numerous target languages, which requires to construct large-scale parallel corpora and introducing substantial training time. In this work, we present an initial exploration of cross-lingual transfer for image-text retrieval tasks in MCLIP models and reduce the high computational time caused by the large-scale parallel corpora.

## 3 Analysis

In this section, we explore influential similarity factors for cross-lingual transferability in multi-modal retrieval scenes, including the identification of potential similarity factors, their measurement methods, and the correlation between these factors and transferability.

### 3.1 Similarity Factors

We adopt token vocabulary overlap to measure token similarity between languages. The token overlap degree $O_{T_k}$ between the source language $S$ and the target language $T_k$ is defined as:

$$O_{T_k} = \frac{|V^S \cap V^{T_k}|}{|V^S \cup V^{T_k}|} \tag{1}$$

where $V^S$ denotes the source language token vocabulary, and $V^{T_k}$ represents the target language token vocabulary. Furthermore, we calculate syntactic similarity ($S_{syn}$), phonological similarity ($S_{pho}$), and geographic similarity ($S_{geo}$) between the source language and the target language using the linguistic features of the URIEL project [27], following Lauscher et al. [28].

### 3.2 Analysis Methodology

We fine-tune the MCLIP model using only the image-text dataset of the source language $\mathcal{D}$. The model is then directly evaluated on target language image-text datasets to measure cross-lingual retrieval performance. We calculate the performance change $\Delta T_k$ for each language before and after fine-tuning and analyze the impact of similarity factors on cross-lingual transferability through visualization techniques and Pearson correlation. To avoid the influence of other factors such as the different language capabilities of pre-training models in different languages [15, 16], we only consider languages which can be well perceived by the pre-trained MCLIP model, resulting in 10 languages[1].

---

[1]The 10 languages are Greek, Spanish, French, Italian, Polish, Portuguese, Swedish, Ukrainian, Chinese-Simplified.

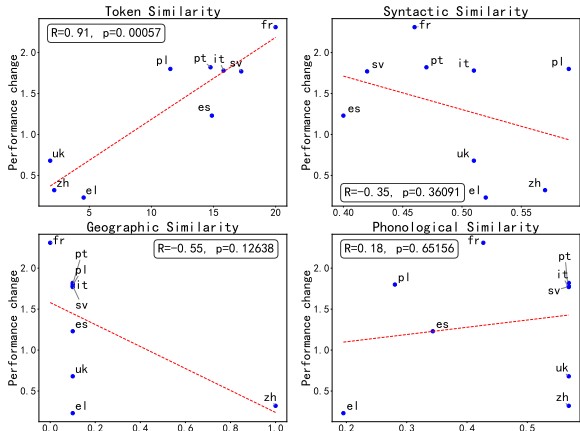

Figure 2: The relationship between the four types of similarity and cross-lingual transferability, Pearson correlation coefficient and p-value is shown in the box.

### 3.3 Analysis Results

As shown in Figure 2, we observe that token similarity significantly impacts cross-lingual transferability, and the p-value of the Pearson correlation test is much lower than 0.05, indicating a strong positive correlation between token similarity and cross-lingual transferability. In contrast, syntactic similarity ($S_{syn}$), phonological similarity ($S_{pho}$), and geographic similarity ($S_{geo}$) do not exhibit a significant influence on cross-lingual transferability. This demonstrates that token similarity is a critical similarity factor that influences cross-lingual transfer performance.

## 4 Methods

In this section, we present our proposed efficient multilingual cross-modal retrieval fine-tuning strategies, including the Transferability-aware Parallel Corpora Learning (TaPCL) and the Critical-information Parallel Corpora Learning (CiPCL) strategies, the overview of our strategies is shown in Figure 3.

### 4.1 Problem Formulation

We formulate multilingual cross-modal retrieval as follows. Given a source language $S$ and $M$ target languages $\{T_k\}_{k=1}^{M}$, we aim to adapt a multilingual vision-language model to downstream tasks using: manually annotated source language image-text pairs $\mathcal{D}^S = \{(v_i, s_i^S)\}_{i=1}^{N}$, where $v_i \in \mathcal{V}$ is an image and $s_i^S \in \mathcal{C}^S$ is text description and machine-translated target language corpora $\mathcal{C}^{T_k} = \{s_i^{T_k}\}_{i=1}^{N}$ generated from $\mathcal{C}^S = \{s_i^S\}_{i=1}^{N}$. The target language training set is defined as:

$$\mathcal{D}^T = \bigcup_{k=1}^{M} \mathcal{D}^{T_k}, \tag{2}$$

where each $\mathcal{D}^{T_k} = \{(v_i, s_i^{T_k})\}_{i=1}^{N}$ contains target language parallel corpora and image pairs.

### 4.2 Transferability-aware Parallel Corpora Learning

Conventional parallel corpora learning approaches [11, 12, 13] uniformly sample all target language training data. Given the target language training set $\mathcal{D}^T$, each sample $(v_i, s_i^{T_k})$ is sampled in mini-batch with equal probability:

$$P_{\text{base}}((v_i, s_i^{T_k}) \in \mathcal{B}) = \frac{B}{M \cdot N}, \tag{3}$$

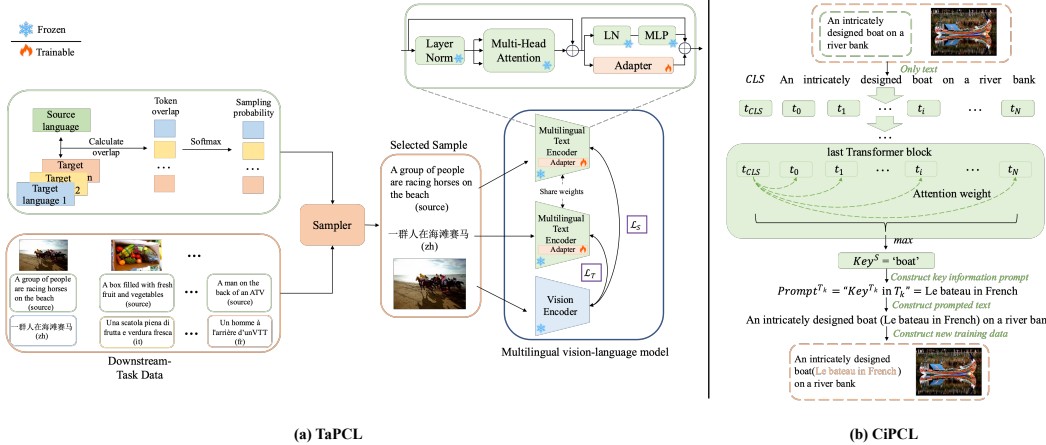

Figure 3: Illustration of our methods: (a) The overview of TaPCL. (b) The overview of CiPCL.

where $\mathcal{B}$ is the mini-batch and $B$ is the batch size. This uniform treatment assumes equitable utilization of parallel corpora across languages.

However, as discussed in Section 3.1, languages with high token overlap require less adaptation effort. To improve fine-tuning efficiency, we propose to reduce the repeated participation of redundant target language samples that are similar to the source language in training by decreasing their sampling probabilities. Specifically, based on the token similarity factor $O_{T_k}$ calculated by Eq. 1, the Transferability-aware Weighted Sampling (taws) function $P_{\text{taws}}$ is formulated as follows:

$$P_{\text{taws}}((v_i, s_i^{T_k}) \in \mathcal{B}) = \frac{B}{N} \cdot \frac{\exp(-O_{T_k}/\tau)}{\sum_{j=1}^{M} \exp(-O_{T_j}/\tau)}, \tag{4}$$

where $\tau$ is a temperature hyperparameter controlling the sharpness of the probability distribution.

By replacing $P_{base}$ with $P_{\text{taws}}$, target language data $\mathcal{D}^{T_k}$ with fewer overlaps with the source language will be assigned a higher sampling probability, while those with large overlaps with the source language will be assigned a lower sampling probability. In this way, our approach encourages the model to focus more on adapting to low-overlap languages while reducing computational overhead associated with high-overlap languages, thereby accelerating downstream task adaptation and lowering the overuse of parallel corpora.

The overall loss combines source and target language objectives through adaptive weighting:

$$\mathcal{L} = \mathcal{L}_S + \alpha\mathcal{L}_T, \tag{5}$$

where $\mathcal{L}_S$ implements image-text alignment in source language and $\mathcal{L}_T$ implements weighted sampling fine-tuning in target language. Both terms use the CLIP contrastive loss:

$$\mathcal{L}_S = \frac{1}{|\mathcal{D}^S|} \sum_{(v_i, s_i^S) \in \mathcal{D}^S} \mathcal{L}_{\text{NCE}}(v_i, s_i^S), \tag{6}$$

$$\mathcal{L}_T = \mathbb{E}_{(v_i, s_i^{T_k}) \sim P_{\text{taws}}} \left[ \mathcal{L}_{\text{NCE}}(v_i, s_i^{T_k}) \right], \tag{7}$$

where $\mathcal{L}_{\text{NCE}}$ denotes the Noise-Contrastive Estimation loss [29]. Based on increasing the sampling probability of samples that are dissimilar from the source language and reducing the sampling probability of similar samples, our optimization objective pays more attention to difficult samples, thereby improving the training efficiency.

## 4.3 Critical-information Parallel Corpora Learning

To further improve fine-tuning efficiency, we discard the full parallel corpus in the target language and instead use only source-language samples with key prompts in the target language for fine-tuning,

thereby avoiding redundant training on semantically similar samples. Specifically, for a source language text $s^S$, instead of constructing complete parallel sentences in target languages, we translate only the critical top-k word $\text{key}^S$ identified in $s^S$ into target languages $\text{key}^{T_k}$. The critical word selection is based on attention weights between the [cls] token and other tokens in the last transformer layer:

$$\text{key}^S = argmax_{t_i \in s, t_i != t_{[cls]}} \sum_{h=1}^{H} \alpha_h^{\text{att}}(t_i, t_{[cls]}), \tag{8}$$

where $H$ denotes the number of attention heads, and $\alpha_h^{\text{att}}$ represents the attention weight between token $t_i$ and the [cls] token in head $h$. If the selected $\text{key}^S$ is a subword unit, we merge it with adjacent tokens until form a complete word. We construct key information prompts using the template:

$$\text{Prompt}^{T_k} = \text{“Key}^{T_k} \text{ in } T_k\text{”},$$

where $T_k$ specifies the target language (e.g., Chinese or French). As shown in Figure 3, the prompt is inserted into the text in the source language, yielding the prompted text $s^{(P,T_k)}$, formally,

$$s^{(P,T_k)} = (t_1, ..., t_{\text{pos}_{\text{key}}}, (\text{Key}^{T_k} \text{ in } T_k), t_{\text{pos}_{\text{key}}+1}, ..., t_n). \tag{9}$$

The training data is sampled through a hybrid strategy:

$$\mathcal{D}^P = \{(v_i, \tilde{s}_i)\}, \tag{10}$$

where

$$\tilde{s}_i = \begin{cases} s_i^S & \text{with prob } c\% \\ s_i^{(P,T_k)} & \text{with prob } (1 - c\%) P_{\text{taws}}(T_k) \end{cases}. \tag{11}$$

To maintain source-language transferability, we randomly retain $c\%$ of source-language texts without prompts. The training objective becomes:

$$\mathcal{L} = \frac{1}{|\mathcal{D}^P|} \sum_{(v_i, \tilde{s}_i) \in \mathcal{D}^P} \mathcal{L}_{\text{NCE}}(v_i, \tilde{s}_i). \tag{12}$$

By fine-tuning without using complete parallel sentences in target languages, we eliminate the dependency on parallel corpora in target languages. This strategy significantly improves computational efficiency while maintaining multilingual generalization. We use the parameter-efficient learning approach Adapter [30] to further improve fine-tuning efficiency.

## 5 Experiments

### 5.1 Experimental Settings

**Datasets and Baseline.** We evaluate our proposed methods on MSCOCO$_{36}$ [10] and XM3600 [31], both of which are cross-modal retrieval benchmarks involving 36 parallel corpora. The pre-trained MCLIP models usually have imbalanced abilities across different languages. As shown in [15, 16], the original ability of the pre-trained multilingual language model will affect the performance of the model in the corresponding language after fine-tuning. As discussed in Section 3, we only consider 10 languages which can be well perceived by the pre-trained OpenCLIP model in our experiments (experiments considering all the target languages are also reported in Table 5 and Figure 7 of the appendix). For MSCOCO$_{36}$ and XM3600, we respectively randomly select 5000 and 2880 samples for training and set English as the source language. For baseline comparisons, we consider: (1) methods that only use the source language to fine-tune, (2) existing cross-lingual transfer approaches (prompt-tuning [23], VME [32]) that do not leverage parallel corpora, and (3) the recent parallel corpora learning method PCL-base [11, 12, 13] which applies uniform sampling across target languages.

**Evaluation Metrics.** Our experiments consider both retrieval performance and training efficiency. To this end, we use rank-based metrics for performance evaluation: Recall@K ($R@K$) for $K = 1, 5, 10$ and $R@Avg$. Specifically, $R@K$ measures the proportion of samples where the correct

Table 1: Performance comparison of multilingual image-text retrieval and number of iterations on MSCOCO$_{36}$ and XM3600.

| | Method | Image-to-Text | | | Text-to-Image | | | R@Avg | Iteration | Runtime |
|---|---|---|---|---|---|---|---|---|---|---|
| | | R@1 | R@5 | R@10 | R@1 | R@5 | R@10 | | | |
| MSCOCO$_{36}$ | Source-only | 50.90 | 80.73 | 90.70 | 49.30 | 79.19 | 89.16 | 73.33 | - | - |
| | Prompt [23] | 52.71 | 81.72 | 90.54 | 49.73 | 78.31 | 88.68 | 73.61 | - | - |
| | VME [32] | 50.75 | 80.16 | 80.91 | 51.24 | 79.92 | 89.83 | 73.62 | - | - |
| | PCL-base [11, 12, 13] | 53.02 | 82.75 | 91.68 | 51.71 | 81.48 | 91.05 | 75.28 | 7039 | 21m02s |
| | TaPCL (Ours) | 54.17 | 82.94 | 91.66 | 51.61 | 81.23 | 90.72 | 75.39 | 4929 | 14m31s |
| | CiPCL (Ours) | 54.17 | 83.17 | 91.64 | 50.68 | 80.87 | 90.38 | 75.15 | 3519 | 09m55s |
| XM3600 | Source-only | 75.03 | 93.25 | 96.66 | 71.29 | 91.10 | 94.87 | 87.03 | - | - |
| | Prompt [23] | 75.17 | 93.42 | 96.81 | 71.46 | 91.25 | 95.03 | 87.19 | - | - |
| | VME [32] | 74.00 | 93.24 | 96.38 | 71.97 | 91.96 | 95.78 | 87.22 | - | - |
| | PCL-base [11, 12, 13] | 76.21 | 94.53 | 97.26 | 72.86 | 92.49 | 96.12 | 88.25 | 8100 | 24m42s |
| | TaPCL (Ours) | 76.32 | 94.54 | 97.29 | 72.93 | 92.46 | 96.08 | 88.27 | 5680 | 16m49s |
| | CiPCL (Ours) | 76.08 | 94.47 | 97.26 | 72.72 | 92.29 | 96.11 | 88.16 | 4050 | 12m05s |

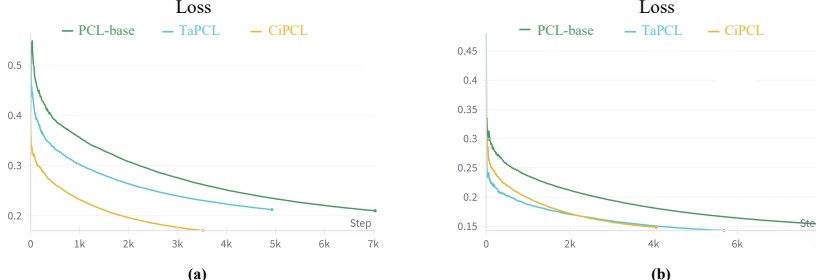

Figure 4: The loss convergence curve on MSCOCO$_{36}$ and XM3600.

image/text is retrieved within the top-$K$ ranked candidates, while $R@Avg$ is the average of text-to-image and image-to-text retrieval $R@K$. Higher values indicate better performance. We report the average metrics across all languages:

$$R@K = \frac{1}{M} \sum_{l=1}^{M} (R@K_l) \tag{13}$$

where $M$ is the number of languages, and $R@K_l$ is the $R@K$ of the $l$-th language. For training efficiency, we consider the training iteration number (with each method utilizing the same batch size) and computation cost until training convergence.

**Implementation Details.** We adopt the OpenCLIP ViT-B-32-XLM-Roberta-Base model [5] as our pre-trained multi-lingual CLIP model. For CiPCL, we use M2M100-1.2B [26] to translate, and we translate the critical top-1 word (we also report the detailed results with the top-2 and top-3 words in A.4.) into target languages. We insert the adapter into the last layer in the model (we also report the detailed results under different trainable parameter numbers in A.2.). For the adapter, we set the up layer output dimension to 256 and the down layer output dimension to 768, and set the dropout probability of the ReLU activation function to 0.1. The weights of the upper layer are initialized with a uniform distribution, while the remaining parameters of the adapter are initialized as zero. The experiments are conducted on Linux with NVIDIA 3090 GPUs, using AdamW optimizer with cosine learning rate scheduler, and the initial learning rate is $1e-4$. We set the batch size to 128 and $\alpha = 0.2$, $\beta = 0.8$. The model is trained for 10 epochs with PCL-base on MSCOCO$_{36}$, and for 20 epochs on XM3600.

## 5.2 Main Results

To validate the multilingual image-text retrieval capability of our fine-tuning strategies, we conduct experiments on MSCOCO$_{36}$ and XM3600. As shown in Table 1, our fine-tuning strategies achieve comparable $R@K_{avg}$ performance to PCL-base, demonstrating equivalent retrieval capabilities.

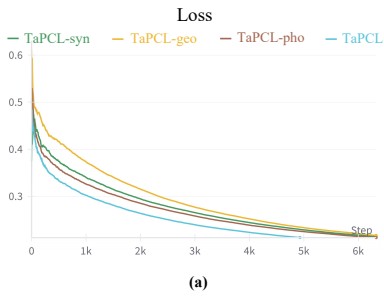
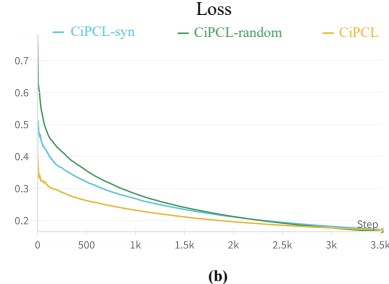

(a)    (b)

Figure 5: The loss convergence curve of ablation study.

In particular, our methods require fewer training iterations than the PCL-base. Specifically, for MSCOCO$_{36}$, TaPCL outperforms PCL-base by 0.11% with reduced training iterations, and CiPCL shows an acceptable performance degradation of 0.13% compared to PCL-base, offset by 50% fewer training iterations. And for XM3600, TaPCL outperforms PCL-base by 0.01%, and CiPCL shows less performance degradation of 0.09%. These results validate that our strategies maintain downstream task performance while significantly improving training efficiency. We also report the detailed results on each target language in Table 9.

To verify the training efficiency of our fine-tuning strategies, we plot loss convergence curves of the three methods on MSCOCO$_{36}$ and XM3600. As shown in Table 1 and Figure 4, compared to PCL-base requiring over 7,000 iterations for loss convergence on MSCOCO$_{36}$ and 8100 iterations on XM3600, TaPCL reaches loss convergence with only 70% of PCL-base iterations and runtime[2]. CiPCL achieves a significantly lower loss than PCL-base with only 50% of its iterations and runtime. This shows that both proposed strategies can achieve stable loss convergence with fewer training iterations and runtime, demonstrating their superior capability to adapt to downstream tasks with reduced computational costs.

## 5.3 Ablation Study

**Ablation Study of TaPCL.** For TaPCL, we explore whether the improved fine-tuning efficiency originates from using token similarity as the primary criterion for adjusting sampling probabilities. We fine-tune the model by separately adopting syntactic similarity, geographic similarity, and phonological similarity as the alternative primary criterion for adjusting sampling probabilities. As shown in Table 2 and Figure 5(a),

Table 2: Ablation study of TaPCL on MSCOCO$_{36}$.

| $S_{syn}$ | $S_{pho}$ | $S_{geo}$ | $O_{T_k}$ | R@Avg | Iteration | Runtime |
|---|---|---|---|---|---|---|
| | | | | 75.28 | 7039 | 21m20s |
| ✓ | | | | 75.12 | 6329 | 19m12s |
| | ✓ | | | 75.26 | 6350 | 19m26s |
| | | ✓ | | 75.01 | 6343 | 19m14s |
| | | | ✓ | 75.39 | 4929 | 14m31s |

we observe that the fine-tuning convergence speeds using other similarity factors as the primary criterion are significantly slower than TaPCL, which shows that the efficiency improvement of TaPCL is specifically attributed to using token similarity as the primary criterion.

**Ablation Study of CiPCL.** For CiPCL, to verify the effectiveness of critical information prompts, we randomly select words from samples of the source language and translate them into target languages as prompts. The model is then fine-tuned using samples augmented with these random prompts. As shown in Table 3 and

Table 3: Ablation study of CiPCL on MSCOCO$_{36}$.

| $S_{syn}$ | $O_{T_k}$ | key$_{T_k}$ | R@Avg | Iteration | Runtime |
|---|---|---|---|---|---|
| | | | 75.28 | 7039 | 21m20s |
| | ✓ | | 74.15 | 3560 | 10m07s |
| ✓ | | ✓ | 74.43 | 3594 | 10m25s |
| | ✓ | ✓ | 75.15 | 3519 | 09m55s |

Figure 5(b), when non-critical information is used as prompts, the retrieval performance decreases significantly, indicating that randomly selected prompts fail to facilitate the learning of visual knowledge for target languages. Similarly to TaPCL, we test CiPCL using syntactic similarity as the primary criterion for adjusting sampling probabilities. The result shows a significant decline in retrieval performance, further validating the effectiveness of using token similarity as the primary criterion.

---

[2]The initial loss difference is caused by the sampling strategies differ across methods; thus, the first mini-batch contains different samples, leading to a slight difference in initial losses. By comparing the early-stage slopes of the loss curves across methods, we can assess the convergence speed.

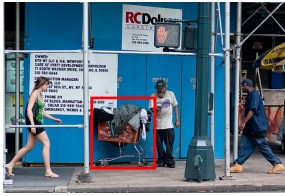 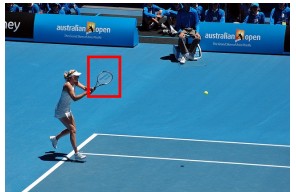 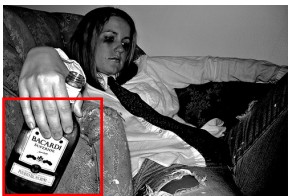

A person with a shopping cart on a city street

A woman standing on a tennis court holding a racquet

A girl with running mascara and a bottle of Bacardi

Figure 6: Sample cases for critical-information selection

## 5.4 Low-resource Languages Analysis

Based on the analysis in Section 3, we further investigate the factors that influence cross-lingual transferability on low-resource languages. Following the categorization proposed by Pratik Joshi [33], we select several low-resource languages and evaluate the Source-only method on them, and calculate the Pearson correlation between token similarity and performance variation, as shown in Table 4. We observe that token similarity still has a significant impact on performance gains—languages with higher token overlap tend to achieve larger improvements. In addition, token similarity remains strongly and positively correlated with performance variation.

Table 4: Correlation between token similarity and performance variation in low-resource languages.

|                  | Danish | Norwegian | Romanian | Quechua | Swahili | Indonesian | Thai | Hebrew | Ukrainian |
|------------------|--------|-----------|----------|---------|---------|------------|------|--------|-----------|
| Token similarity | 0.19   | 0.17      | 0.16     | 0.16    | 0.14    | 0.14       | 0.03 | 0.01   | 0.02      |
| R@Avg variation  | 1.89   | 1.77      | 1.08     | 0.99    | 1.05    | 1.02       | 0.33 | 0.23   | 0.68      |
| Correlation      | R = 0.84, P = 0.004 |||||||||

## 5.5 Critical-information Modeling

Figure 6 shows sample cases for critical-information selection. We can see that the selected critical information (e.g., the shopping cart) carries the key visual knowledge about the corresponding image, supporting our claim for our proposed CiPCL method.

## 6 Conclusion and Limitations

This work focuses on MCLIP models and analyzes the similarity factors that influence cross-lingual transfer in image-text retrieval tasks. Based on the analysis, we propose two efficient parallel corpus learning strategies to alleviate the high computational overhead caused by directly incorporating all parallel corpora. Experiments on two multilingual image-text retrieval datasets demonstrate that the proposed methods enhance the efficiency of parallel corpus-based fine-tuning for downstream tasks.

TaPCL and CiPCL demonstrate high efficiency when fine-tuning for cross-modal retrieval tasks, suggesting their potential scalability. In our future work, we will not only focus on coarse-grained multilingual retrieval tasks but will further investigate the application of TaPCL and CiPCL in finer-grained vision-language tasks, such as multilingual VQA.

## Acknowledgments and Disclosure of Funding

This work was supported by the Natural Science Foundation of Sichuan (No. 2025YFHZ0124), the Sichuan Science and Technology Program (No. 2024NSFTD0036), the Frontier Cross Innovation Team Project of Southwest Jiaotong University (YH1500112432297), and the Engineering Research Center of Sustainable Urban Intelligent Transportation, Ministry of Education.

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

# A  Additional Results

In this section, we report findings from the following experiments: (1) multilingual fine-tuning with the expanded language set, (2) multilingual fine-tuning under scalable trainable parameters, (3) unimodal multilingual model adaptation, and (4) critical information quantity effects in CiPCL. All experiments are conducted on $MSCOCO_{36}$.

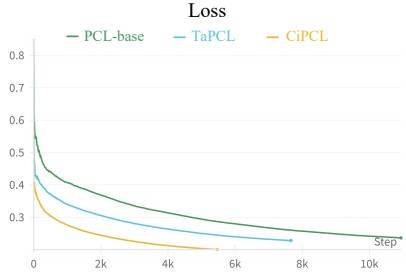

Figure 7: The loss convergence curve on $MSCOCO_{36}$ in 29 languages.

## A.1  Results with All the Target Languages Considered

To further validate the effectiveness of our method when applied to a wider range of languages, we expand the number of languages and exclude languages not involved in the pre-training phase of the MCLIP model, resulting in 29 languages[3]. As shown in Table 5 and Figure 7, our method maintains strong retrieval performance with fewer training iterations even when scaling to more languages. The convergence speeds of TaPCL and CiPCL are not slowed by the increased number of languages, remaining faster than the PCL-base. This demonstrates that the effectiveness of our method is not constrained by language quantity and consistently enhances fine-tuning efficiency when handling multilingual scenarios.

Table 5: Performance of multilingual image-text retrieval and number of iterations on $MSCOCO_{36}$ in 29 languages.

| Method | R@Avg | Iteration | Runtime |
|---|---|---|---|
| PCL-base | 72.73 | 10940 | 36m02s |
| TaPCL | 73.13 | 7660 | 25m11s |
| CiPCL | 72.51 | 5470 | 17m45s |

## A.2  Scalability under Different Trainable Parameter Numbers

To verify the effectiveness of our method under varying quantities of learnable parameters, we configure adapters in different transformer layers. We set $i = 8$, 9, and 10, and insert adapters into the layer $l$ where $l \geq i$, followed by model fine-tuning. As shown in Table 6 and Figure 8, our method consistently improves fine-tuning efficiency across different adapter configurations, demonstrating that our fine-tuning strategy remains effective regardless of the number of learnable parameters.

Table 6: Performance of multilingual image-text retrieval and number of iterations on $MSCOCO_{36}$ with different amounts of adapters.

| Method | Layer >= 10 | | | Layer >= 9 | | | Layer >= 8 | | |
|---|---|---|---|---|---|---|---|---|---|
| | R@Avg | Iteration | Runtime | R@Avg | Iteration | Runtime | R@Avg | Iteration | Runtime |
| PCL-base | 74.31 | 7039 | 22m25s | 74.53 | 7039 | 23m27s | 74.70 | 7039 | 24m31s |
| TaPCL | 74.51 | 4929 | 15m22s | 74.72 | 4929 | 16m06s | 74.94 | 4929 | 17m01s |
| CiPCL | 74.39 | 3519 | 11m05s | 74.52 | 3519 | 11m42s | 74.46 | 3519 | 12m08s |

---

[3]The 29 languages are Arabic, Czech, Danish, German, Greek, Spanish, Persian, Finnish, French, Croatian, Hungarian, Indonesian, Italian, Hebrew, Japanese, Korean, Dutch, Norwegian, Polish, Portuguese, Romanian, Russian, Swedish, Thai, Turkish, Ukrainian, Vietnamese, Chinese-Simplified.

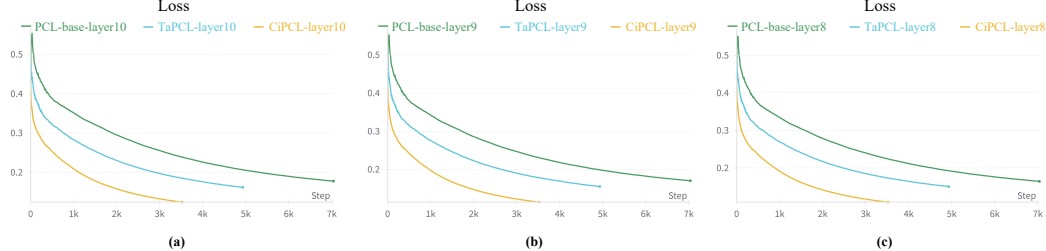

Figure 8: The loss convergence curve on $MSCOCO_{36}$ with different amounts of adapters.

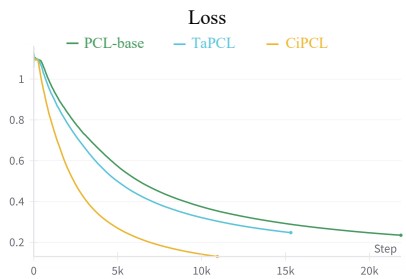

Figure 9: The loss convergence on XNLI.

## A.3 Scalability to Unimodal Multilingual Model

To validate the portability of TaPCL and CiPCL, we apply them to a unimodal fine-tuning scenario. We fine-tune the multilingual pre-trained model XLM-RoBERTa-base [21] on the XNLI dataset [11], which covers 15 languages, with English as the source language and target language training sets derived from English translations. Following Wu et al. [14], we adopted full

Table 7: Performance of natural language inference and number of iterations on XNLI.

| Method | Accuracy | Iteration | Runtime |
|---|---|---|---|
| PCL-base | 59.97 | 21874 | 1h17m |
| TaPCL | 60.46 | 15314 | 0h52m |
| CiPCL | 59.84 | 10939 | 0h38m |

fine-tuning while freezing the parameters of the first 6 layers. As shown in Table 7 and Figure 9, our methods significantly improve fine-tuning efficiency when applied to the unimodal multilingual model, demonstrating their effectiveness even when applied to single-modal scenarios.

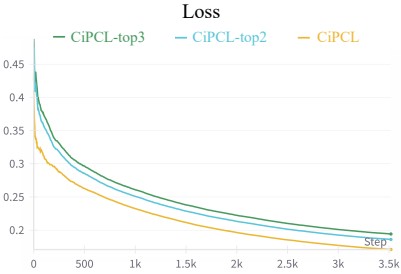

Figure 10: The loss convergence with different quantities of critical information prompts.

## A.4 Effects of Critical-information Quantity

This work further investigates the impact of the number of critical information on CiPCL. Specifically, we select tokens with top-k attention weights relative to the [CLS] token as key words, where k=2,3. Subsequently, we construct key information prompts using the selected keywords. As presented in Table 8 and Figure 10, we observe that neither retrieval performance nor fine-tuning efficiency

improved further despite an increase in the number of key information prompts. We speculate that when excessive key information prompts are introduced, these prompts may function as noise relative to the original English samples. This noise effect potentially creates a counterbalance where the positive influence of key information prompting is offset by the diminishing effectiveness of cross-lingual transfer.

Table 8: Performance of multilingual image-text retrieval and number of iterations on MSCOCO$_{36}$ with different quantities of critical information prompts.

| Method | R@Avg | Iteration | Runtime |
|---|---|---|---|
| CiPCL-top-3 | 75.05 | 3519 | 10m46s |
| CiPCL-top-2 | 75.01 | 3519 | 10m22s |
| CiPCL-top-1 | 75.15 | 3519 | 09m55s |

## A.5 Results on Each Target Language

Table 9 reports the detailed results on each target language. We can see that our proposed approaches can obtain competitive performance on each target lanagueg compared to the PCL-base model which directly utilizes the parallel corpus.

Table 9: Performance of multilingual image-text retrieval and number of iterations on MSCOCO$_{36}$ with different quantities of critical information prompts.

| Method | English | Greek | Spanish | French | Italian | Polish | Portuguese | Swedish | Ukrainian | Chinese |
|---|---|---|---|---|---|---|---|---|---|---|
| Source-only | 78.87 | 71.73 | 73.33 | 74.30 | 74.37 | 71.83 | 73.88 | 71.87 | 70.88 | 72.23 |
| PCL-base | 78.63 | 74.45 | 76.02 | 75.92 | 76.05 | 74.57 | 75.78 | 74.72 | 72.83 | 73.85 |
| TaPCL | 78.70 | 74.63 | 76.47 | 75.37 | 76.40 | 75.17 | 75.93 | 73.52 | 73.12 | 74.58 |
| CiPCL | 79.43 | 74.13 | 75.98 | 75.73 | 76.23 | 74.33 | 75.43 | 74.00 | 72.33 | 73.90 |

