# OpenReview forum: "Efficiently Maintaining the Multilingual Capacity of MCLIP in Downstream Cross-Modal Retrieval Tasks"
_NeurIPS.cc/2025/Conference — NeurIPS 2025 poster_

### Official Review · Reviewer_BWpp · 2025-06-30

**Clarity:** 3
**Significance:** 2
**Originality:** 3
**Rating:** 4
**Confidence:** 4

**Summary:**

The paper proposes TaPCL and CiPCL strategies to address the problem of multilingual capability degradation during fine-tuning of Multilingual CLIP (MCLIP). TaPCL reduces redundant computation by reducing the sampling probability of languages with high token similarity, and CiPCL achieves precise knowledge transfer by translating key terms in the source language and inserting prompts. The experiments were verified on the MSCOCO₃₆ and XM3600 datasets, providing a new path for MCLIP to balance computational efficiency and multilingual robustness.

**Questions:**

1. After the data set is selected and processed, the amount of data is small, and the evaluation of the model's true capabilities may not be very accurate. There is also a risk of overfitting.

2. Token similarity and key information extraction strategies may have certain limitations.

For example, token overlap cannot distinguish the semantic differences of polysemous words, such as the English word "bank" and the German word "Bank" (bench) are similar in form but unrelated in meaning. TaPCL may misjudge them as high-similarity language pairs, resulting in a lower sampling probability.

CiPCL only extracts the top-1 key terms and ignores the semantic associations of multiple term combinations. For example, "artificial intelligence" needs to be translated into both "artificial" and "intelligence" to fully convey the semantics.

3. Are all batch sizes set to the same? Because for retrieval tasks, the batch size also has a significant impact on the convergence speed and effect.

4. Are the initial losses of different experimental settings in Figures 4 and 5 consistent? It seems inconsistent. Why?

5. This paper only verifies the effectiveness of the experiment through cross-modal retrieval, and the generalization of multi-tasks is questionable, especially with such a small amount of training data.

**Ethical Concerns:**

["NO or VERY MINOR ethics concerns only"]

**Final Justification:**

I thank the author for their response, which addresses my concerns. I will raise my score.

**Limitations:**

yes

**Quality:**

2

**Strengths And Weaknesses:**

Strengths:

Dynamically optimize sampling based on token similarity to avoid redundant training of highly similar languages, reducing the number of iterations by 30%-50%.

Only key terms are translated and inserted into the source corpus, without the need for full parallel data, reducing the amount of data while maintaining performance.

**Weaknesses:**

1. After the data set is selected and processed, the amount of data is small, and the evaluation of the model's true capabilities may not be very accurate. There is also a risk of overfitting.

2. Token similarity and key information extraction strategies may have certain limitations.

For example, token overlap cannot distinguish the semantic differences of polysemous words, such as the English word "bank" and the German word "Bank" (bench) are similar in form but unrelated in meaning. TaPCL may misjudge them as high-similarity language pairs, resulting in a lower sampling probability.

CiPCL only extracts the top-1 key terms and ignores the semantic associations of multiple term combinations. For example, "artificial intelligence" needs to be translated into both "artificial" and "intelligence" to fully convey the semantics.

3. Are all batch sizes set to the same? Because for retrieval tasks, the batch size also has a significant impact on the convergence speed and effect.

4. Are the initial losses of different experimental settings in Figures 4 and 5 consistent? It seems inconsistent. Why?

5. This paper only verifies the effectiveness of the experiment through cross-modal retrieval, and the generalization of multi-tasks is questionable, especially with such a small amount of training data.

---

> ### Author Rebuttal · Authors · 2025-07-31
>
> **Q1: The dataset is small, risking overfitting and unreliable evaluation.**
>
> A1: Thank you for this question. Our data scale is sufficient for training and validating an adapter‑based multilingual CLIP fine‑tuning model. Specifically, on MSCOCO we sample 5,000 instances per language across ten target languages, yielding 50,000 training samples in total, as for XM3600, the total number of training sample is 28,800, which are adequate for robust evaluation. In addition, CiPCL essentially acts as a data augmentation method (injecting key terms into the original sentence), which increases diversity and is less prone to overfitting. Moreover, as shown in Table 1, we also provide an explicit overfitting check. The performance gap between the training and test sets is consistently ≤ 1.5%, indicating no memorization effects. Meanwhile, we adopt the adapter model to fine-tuning MCLIP, which is a parameter-efficient fine-tuning technique. Adapter does not update the original pre-trained parameters but instead adds LoRa residual layers, thus reducing the risk of overfitting. Therefore, our evaluation is reliable and overfitting is minimal. We will discuss this more clearly in the final version.
>
> **Table 1: Performance on training set and test set of MSCOCO$_{36}$**
> |Method|Training set R@Avg|Test set R@Avg|
> |:---|:---:|:---:|
> |PCL-base|76.48|75.28|
> |TaPCL|76.95|75.39|
> |CiPCL|76.56|75.15|
> ||||
>
> **Q2 Token similarity and key term extraction may misinterpret polysemy or ignore multi-word semantics.**
>
> A2: Thank you for this question. In TaPCL, we measure cross‑lingual similarity using the language–level overall token overlap. Therefore, several polysemous cases do not dominate the decision. The role of token overlap in cross‑lingual transfer has been emphasized in prior work [1–3], and our empirical results in Figure 1 corroborate this. For CiPCL, as shown in Table 7 in the appendix, we compared multi‑key‑term and single‑key‑term prompts, where multi‑term prompts bring no significant gains while increasing training time, so we adopt the top‑1 scheme, which maintains effectiveness while reducing overhead. Moreover, we believe that a single key term can still serve as a strong semantic anchor (e.g., artificial → “人工” (Chinese) or “artificielle” (French)). Therefore, our similarity metric and term‑selection strategy are reasonable and effective in practice. We will discuss this point more clearly in the final version.
>
> [1] Telmo Pires, Eva Schlinger, and Dan Garrette. "How Multilingual is Multilingual BERT?" ACL 2019.
>
> [2] Vaidehi Patil, Partha Talukdar, and Sunita Sarawagi. "Overlap-based Vocabulary Generation Improves Cross-lingual Transfer Among Related Languages." ACL 2022.
>
> [3] Tomasz Limisiewicz, Jiří Balhar, and David Mareček. "Tokenization Impacts Multilingual Language Modeling: Assessing Vocabulary Allocation and Overlap Across Languages." ACL Findings 2023.
>
> **Q3: Are batch sizes consistent across experiments?**
>
> A3: Thank you for this question. To ensure a fair comparison, all experiments, including all the baselines and our variants, use the same batch size of 128. Similar results can also be observed based on other batch sizes (e.g., 64 and 256). We will discuss this more clearly in the final version.
>
> **Q4: Why are initial losses in Figures 4 and 5 inconsistent?**
>
> A4: Thank you for this question. The initial loss difference is caused by the sampling strategies differ across methods; thus, the first mini‑batch contains different samples, leading to different starting losses in Figures 4 and 5, but their initial accuracy is the same. By comparing the early‑stage slopes of the loss curves across methods, we can assess convergence speed. From Figures 4 and 5, our approach shows a steeper initial slope, indicating higher convergence efficiency. Moreover, we will add accuracy‑vs‑iteration curves to further validate the effectiveness of the proposed approach in the final version. We will discuss this point more clearly in the final version.
>
> **Q5: Generalization ability for other tasks.**
>
> A5: Thank you for this question. Our study focuses on the fine‑tuning mechanisms of multilingual CLIP. As multilingual cross‑modal retrieval is the primary application scenario for multilingual CLIP, we organize our main experiments on this task. To further assess the scalability of the proposed approach, Appendix A.3 reports results on the multilingual natural language inference task. We find that, on this task as well, our method achieves substantial training‑efficiency gains without loss in accuracy. These results verify the scalability of our approach. We will discuss this more clearly in the final version. We will discuss this more clearly in the final version.

---

> ### Author Response · Authors · 2025-08-06
>
> Dear Reviewer,
>
>      We sincerely appreciate your time and effort in reviewing our manuscript and offering valuable suggestions. Since the discussion DDL comes closer, we would like to confirm whether our responses have effectively addressed your concern. If you require further clarification, please do not hesitate to contact us. We are more than willing to continue the discussion with you.
>
> Warm regards,
>
> Authors

---

### Official Review · Reviewer_aAer · 2025-07-01

**Clarity:** 3
**Significance:** 1
**Originality:** 2
**Rating:** 4
**Confidence:** 2

**Summary:**

This paper addresses the issue that MCLIP requires training data for each language when fine-tuning for downstream tasks. From the perspective of optimizing data, the authors propose two methods: TaPCL, which performs sampling based on language similarity, and CiPCL, which retains key knowledge in the data. Experiments conducted on two cross-model retrieval benchmarks demonstrate the effectiveness of the proposed methods.

**Questions:**

1. Practical Applicability: Under what specific real-world scenarios would it be necessary to fine-tune a model for multiple languages using the proposed methods? Would it not be simpler to fine-tune separate versions of the model for each language and provide these versions to relevant users? Please elaborate on potential use cases or advantages of your approach in comparison to this alternative.

2. Experimental Scope: Why were the experiments limited to only two cross-modal retrieval benchmarks? Is there a specific reason for focusing on these tasks, and do you believe the proposed methods are applicable to other types of downstream tasks? If so, could you provide additional evidence or reasoning to support this?

3. Performance and Impact: The proposed methods yield only marginal performance improvements and, in some cases, slightly underperform compared to PCL-base. Additionally, the time savings of approximately ten minutes may not be impactful in real-world scenarios. Could you provide further explanation or justification regarding these results, and clarify the significance of the observed improvements in practical settings?

**Ethical Concerns:**

["NO or VERY MINOR ethics concerns only"]

**Final Justification:**

Thank you for the rebuttal which cleared up all my confusion. I will adjust my rating to 4 accordingly.

**Limitations:**

yes

**Quality:**

2

**Strengths And Weaknesses:**

## Strengths
1. Reasonable Motivation and Preliminary Evidence: The proposed methods are based on sound motivation and are supported by initial experimental results, which provide some validation for their effectiveness.

2. Clear Introduction: The introduction is well-structured and clearly identifies the gaps in existing work, as well as the positioning of this study. This makes the contribution of the paper easy to understand.

## Weakness
1. Limited Practical Applicability: The real-world applicability of the proposed methods might be constrained. This limitation could be inherent in this type of approach. Specifically, under what circumstances would it be necessary to fine-tune the model for multiple languages? Wouldn't it be simpler to fine-tune a version of the model for each language separately and provide these versions to the relevant users? The authors are encouraged to propose a practical scenario where their methods would be advantageous.

2. Experimental Scope and Task Variety: The experimental setup is limited to two cross-modal retrieval benchmarks. The authors should explain why the experiments were conducted exclusively in this context and whether it would be appropriate to extend the evaluation to other task types.

3. Performance Concerns: The performance improvement brought by the proposed methods is not significant and, in some cases, slightly underperforms compared to PCL-base. While the time savings of approximately ten minutes are noted, such a minor difference in real-world scenarios is unlikely to be impactful. The authors are encouraged to provide further clarification regarding these aspects.

---

> ### Author Rebuttal · Authors · 2025-07-31
>
> **Q1 Why train a single multilingual model instead of multiple independent monolingual models?**
>
> A1: Thank you for this question. Compared with monolingual models, a single multilingual model offers advantages in training efficiency, inference efficiency, and operational simplicity, and therefore has broad real‑world applicability. Specifically, a multilingual model leverages cross‑lingual similarity to enable knowledge transfer across languages, which reduces training cost and improves performance on low‑resource languages. In multimodal retrieval, queries are naturally multilingual or even mixed‑language inputs (e.g., social‑media search). Additional language‑identification and routing stages would be required when deploying separate monolingual models, which struggle with mixed‑language inputs and introduce extra latency and failure modes. Moreover, language‑specific traffic fluctuates over time; maintaining many monolingual models complicates capacity planning and load balancing. A single multilingual model avoids these issues while providing consistent behavior across languages, making it the more practical choice in many real‑world scenes. We will more clearly clarify this in the final version.
>
> **Q2 Can the proposed methods be applied to other tasks?**
>
> A2: Thank you for this question. Our study focuses on the fine‑tuning mechanisms of multilingual CLIP. As multilingual cross‑modal retrieval is the primary application scenario for multilingual CLIP, we organize our main experiments on this task. To further assess the scalability of the proposed approach, Appendix A.3 reports results on the multilingual natural language inference task. We find that, on this task as well, our method achieves substantial training‑efficiency gains without loss in accuracy. These results verify the scalability of our approach. We will discuss this more clearly in the final version.
>
> **Q3 The practical impact of improving training efficiency.**
>
> A3: Thank you for this question. Our method reduces training time by 30%–50% while achieving the same accuracy as standard multilingual‑CLIP fine‑tuning methods (PCL‑base). This gain is proportional rather than a fixed offset. In real‑world settings, fine‑tuning datasets are often very large—for example, the full WIT dataset contains about 170M training samples, and a single fine‑tuning run with PCL‑base on 4 GPUs takes about 600 hours. Under the same setup, our approach could significantly cut training time by 180 to 300 hours. Moreover, in real‑world scenes, models are updated frequently as new data and new businesses emerge; thus the training‑efficiency improvements delivered by our approach have concrete practical value. We will discuss this point more clearly in the final version.

---

> > ### Comment · Reviewer_aAer · 2025-08-04
> >
> > Thank you for the rebuttal which cleared up all my confusion. I will adjust my rating to 4 accordingly.

---

> > > ### Author Response · Authors · 2025-08-05
> > >
> > > Thanks for your positive recognition about our work.

---

### Official Review · Reviewer_Unow · 2025-07-02

**Clarity:** 3
**Significance:** 2
**Originality:** 2
**Rating:** 4
**Confidence:** 3

**Summary:**

# Method summary

The paper focuses on a problem in downstream adaptation of Multilingual CLIP (MCLIP) models - maintaining performance in other languages that was not present in the downstream fine-tuning, which is usually the case without directly using parallel corpora in other languages. The authors first analyse token overlap degree (Otk) between a source and target lanaguage (IOU over the token vocab), syntactic, phonological and geographic similarlity between the MCLIP model before and after fine-tuning on the source language. The analysis shows that only token overlap degree correlates well to drop in performance. The authors then present TaPCL and CiPCL: the idea behind TaPCL is uniform sampling from target languages can be improved since the analysis shows high overlap indicates less adaptation effort and hence propose a weighted sampling based on the token overlap degree. Furthermore, in CiPCL, the authors propose focusing on top-k critical word key in the source lanaguages and only translating those into target languages and similar weighted sampling strategy to TaPCL is used for source and target languages.

# Experiment setup

The authors evaluate on MSCOCO and XM3600.

# Results summary
Table one shows ~1% absolute improvement over the closest baseline on MSCOCO and XM3600 while having a faster runtime and fewer iterations than PCL-base. Ablations suggest all the similarlity measures roughly perform the same and Otk performs the best in CiPCL.

**Questions:**

No additional questions to the authors, just the ones listed in the weaknesses.

**Ethical Concerns:**

["NO or VERY MINOR ethics concerns only"]

**Final Justification:**

> Thank you for the clarification and updated results on the seeds, the relatively lower std deviation boosts my confidence in inferring the efficiency gains while preserving / slightly improving the performance. Bumping score to 4.

**Limitations:**

Yes, the authors have addressed limitations but no mention of the broader impact statement which is clearly a part of the author guidelines.

**Paper Formatting Concerns:**

No mention of the broader impact statement which is clearly a part of the author guidelines.

**Quality:**

2

**Strengths And Weaknesses:**

# Strengths

1. Clearly written paper with well-described problem statement, methodology, and contributions.
2. The analysis and motivations are clear in Section 3

# Weaknesses

1.  The authors do not provide any information on seeds etc. With the improvements being really close to the baseline, I wonder how much can we really infer from a single run.
2. The ablations indicate Otk roughly performs the same - if the correlations were really that bad, the performance changes should also have been reflected the same, however, only runtime / iterations are effected. I'm not sure I see an explanation or hypothesis as to why this happens.

---

> ### Author Rebuttal · Authors · 2025-07-31
>
> **Q1: Reproducibility & statistical reliability.**
>
> A1: Thank you for this question. There may be a misunderstanding on our main focus. As stated in the Introduction, this work primarily targets the fine‑tuning efficiency of multilingual CLIP models rather than maximizing performance. As stated in the Introduction, directly incorporating parallel corpora for fine-tuning will cause substantial computational overhead, hindering the flexible deployment of MCLIP models for downstream applications; thus, it is necessary to improve fine-tuning efficiency. PCL‑base can be regarded as an upper bound on multilingual performance as it uses the full parallel corpus; therefore, our goal is to improve training efficiency while achieving comparable performance with PCL‑base. Regarding random seeds, all experiments in the paper used a fixed, randomly chosen seed (42). To more reliably validate our results, we additionally ran experiments with three seeds (i.e., 43, 44, and 45); the results are reported in Table 1 and Table2. The outcomes are consistent across different seeds, showing performance comparable to PCL‑base along with >30% improvements in training efficiency, which confirms the robustness of our method and the reliability of our conclusions. We will discuss this more clearly in the final version.
>
> **Table 1: Performace on MSCOCO$_{36}$ with different seeds**
> |Method|R@Avg|Iteration|Runtime|Average of R@avg|SD|
> |:---|:---:|:---:|:---:|:---:|:---:|
> |PCL-base seed1|75.30|7036|21m02s|||
> |PCL-base seed2|75.15|7039|21m07s|75.27|0.11|
> |PCL-base seed3|75.36|7035|21m04s|||
> |TaPCL seed1|75.42|4940|14m51s|||
> |TaPCL seed2|75.50|4957|15m06s|75.45|0.04|
> |TaPCL seed3|75.44|4946|14m58s|||
> |CiPCL seed1|75.16|3523|09m59s|||
> |CiPCL seed2|75.10|3518|09m54s|75.16|0.06|
> |CiPCL seed3|75.21|3522|10m01s|||
> |||||||
>
> **Table 2: Performace on XM3600 with different seeds**
> |Method|R@Avg|Iteration|Runtime|Average of R@avg|SD|
> |:---|:---:|:---:|:---:|:---:|:---:|
> |PCL-base seed1|88.22|8100|24m41s|||
> |PCL-base seed2|88.26|8100|24m42s|88.24|0.02|
> |PCL-base seed3|88.23|8110|24m54s|||
> |TaPCL seed1|88.24|5688|16m55s|||
> |TaPCL seed2|88.29|5680|16m50s|88.27|0.03|
> |TaPCL seed3|88.28|5678|16m46s|||
> |CiPCL seed1|88.03|4058|12m14s|||
> |CiPCL seed2|88.10|4044|11m53s|88.06|0.04|
> |CiPCL seed3|88.06|4042|11m52s|||
> |||||||
>
> **Q2: Why weak Otk correlations slow convergence but leave final accuracy essentially unchanged?**
>
> A2: Thank you for this question. As stated in our rebuttal to the above question, our objective is to improve fine‑tuning efficiency, rather than maximize accuracy. When parallel corpora are used, different sampling strategies will, given sufficient iterations, converge towards the same upper‑bound accuracy (comparable to PCL‑base); the difference is how quickly they reach it. Sampling based on other similarity measures leverages source‑language transferability less effectively and thus has lower training efficiency than Otk. Moreover, as shown in Table 3, when training is stopped after the same iteration number, the variants using other sampling methods underperform Otk. We will discuss this in more detail in the final version.
>
> **Table 3: Ablation study of TaPCL under the same number of iterations**
> |Mehtod|R@Avg|Iteration|Runtime|
> |:---|:---:|:---:|:---:|
> |TaPCL-syn|74.21|4929|14m39s|
> |TaPCL-pho|74.42|4929|14m50s|
> |TaPCL-geo|74.17|4929|14m35s|
> |TaPCL-token|75.39|4929|14m31s|
> |||||
>
> **Q3: No mention of the broader impact statement, which is clearly a part of the author guidelines.**
>
> A3: Thank you for this question. As required by the author guidelines, we explained in the checklist why this work does not include a broader impact statement. This work does not present any dataset or benchmark. It proposes strategies to enhance the efficiency of parallel corpus-based fine-tuning for downstream tasks. Therefore, there is no potential societal harm associated with this work. We will more clearly clarify this in the final version.

---

> > ### Comment · Reviewer_Unow · 2025-08-01
> > **Acknowledging**
> >
> > Thank you for the clarification and updated results on the seeds, the relatively lower std deviation boosts my confidence in inferring the efficiency gains while preserving / slightly improving the performance.
> > Bumping score to 4.

---

> > > ### Author Response · Authors · 2025-08-04
> > >
> > > Thanks for the recognition about our work.

---

### Official Review · Reviewer_vHM2 · 2025-07-03

**Clarity:** 3
**Significance:** 2
**Originality:** 3
**Rating:** 5
**Confidence:** 3

**Summary:**

This paper addresses the computational inefficiency of maintaining multilingual capabilities when fine-tuning MCLIP models for downstream tasks. The authors find that token similarity is the key factor affecting cross-lingual transferability in multimodal retrieval. They propose two strategies: TaPCL, which adjusts sampling probabilities based on token similarity, and CiPCL, which augments source text with translated key terms instead of full parallel sentences. Experiments on MSCOCO36 and XM3600 show both methods achieve comparable performance to baselines while reducing training time by 30-50%.

**Questions:**

Do you have results on languages that have high token overlap but usually are poor in digital resources according to this taxonomy [1]?

[1] Joshi, Pratik, et al. "The state and fate of linguistic diversity and inclusion in the NLP world." ACL 2020.

**Ethical Concerns:**

["NO or VERY MINOR ethics concerns only"]

**Final Justification:**

I jumped up my score to 5 because my main concern (about seeing results on the low-resource languages with a lot of token overlap with the English) is addressed.

**Quality:**

3

**Strengths And Weaknesses:**

**Strengths**: The paper provides a systematic empirical analysis to identify token similarity as the primary factor affecting cross-lingual transfer in multimodal settings, which is a novel contribution to the field. The insights about token similarity could inform future work on multilingual multimodal models. Furthermore, I also think that the proposed methods are intuitive and motivated, with clear mathematical formulations for both TaPCL and CiPCL strategies. That said, I am not super familiar with multimodal models literature to know how novel these two strategies are. The wide language coverage also adds to the strengths of the paper.

**Weaknesses**: A critical weakness is that token similarity is inherently confounded with pretrained representation quality - languages with high token overlap typically receive better multilingual pretraining due to shared subwords and data availability. This makes it unclear whether the observed transferability stems from token similarity itself or simply reflects which languages already have stronger representations in the base model. In addition, the authors say that in Section 5 they only consider "10 languages which can be well perceived by the pre-trained MCLIP model" which complicates this fundamental confound. To disentangle this, it'd be encouraging to see results on languages that have low digital resources but have a lot of token overlap with English.

Furthermore, in my opinion, the paper doesn't adequately discuss failure cases or provide guidance on when to use TaPCL versus CiPCL. Scalability concerns can arise from the need to compute pairwise token similarities and the additional translation model requirement for CiPCL. The key term selection in CiPCL (Equation 8) seems simplistic - using only attention weights might miss important contextual information (such as knowledge retrieval pairs [1]). However, I don't think this is a fundamental weakness.


[1] Geva, Mor, et al. "Transformer feed-forward layers are key-value memories." EMNLP 2021.

---

> ### Author Rebuttal · Authors · 2025-07-31
>
> **Q1: Are the improvements from token similarity, or from uneven base‑model language quality?**
>
> A1: Thank you for the question. In this work, we selected 9 target languages with comparable zero‑shot performance to more clearly  isolate the effect of token similarity on transferability. This design effectively removes the confounding influence of base‑model pretraining quality on transfer. The zero‑shot results for these 9 target languages are reported in Table 1, which shows that the base model has similar zero-shot capability on those target languages. Moreover, our conclusions also hold for low‑resource languages: following your suggestion, we evaluated the Source-only method (fine‑tuned only on the source language) on several low‑resource languages from [1], with results in Table 2. We observe that token overlap still has a significant effect on performance gains—languages with higher token overlap achieve larger improvements. We will make this more explicit in the final version.
>
> [1] Pratik Joshi, Sebastin Santy, et al. "The state and fate of linguistic diversity and inclusion in the NLP world." ACL 2020.
>
> **Table 1: Zeroshot performance on 9 target languages**
> |Language|Greek|Spanish|French|Italian|Polish|Portuguese|Swedish|Ukrainian|Chinese|
> |:------:|:----:|:------:|:------:|:-----:|:------:|:--------:|:-------:|:--------:|:-------:|
> |Zeroshot|71.50|72.10|71.99|72.59|70.03|72.06|70.12|70.20|71.91|
> ||||||||||
>
> **Table 2: Performance of low digital resource languages**
> |Language|Performance change|Zeroshot|After fine-tuning|Token overlap %|
> |:---|:---:|:---:|:---:|:---:|
> |Danish|1.89|65.88|67.77|18.92|
> |Norwegian|1.77|63.98|65.75|17.17|
> |Romanian|1.08|55.05|56.13|16.03|
> |Quechua|0.99|28.23|29.22|15.87|
> |Swahili|1.05|13.38|14.43|14.39|
> |Indonesian|1.02|66.55|67.57|13.92|
> |Thai|0.33|55.93|56.26|2.67|
> |Hebrew|0.23|66.08|66.31|1.45|
> |Ukrainian|0.68|70.20|70.88|1.85|
> |**R=0.84 P=0.004**|||||
> ||||||
>
> **Q2: When to use TaPCL vs. CiPCL?**
>
> A2: Thank you for the question. TaPCL and CiPCL can be chosen as needed. Empirically, TaPCL achieves slightly higher accuracy than CiPCL (+0.24% on MSCOCO, +0.11% on XM3600), but requires more fine‑tuning time (+46% on MSCOCO, +39% on XM3600). We therefore recommend selecting between them by trading off accuracy and training time for the target use case. We will make this guidance explicit in the final version.
>
> **Q3: Scalability and dependence on computing token similarities and translation model.**
>
> A3: Thank you for the question. As target-language training samples are hard to collect, the target-language training data in MSCOCO and XM3600 are obtained by translating source‑language datasets. Under this setup, the use of a translation model aligns with the existing pipeline. As for token similarity computation, since it only uses input-layer token embeddings and is performed once as a pre-training step, computing token similarities for the entire dataset takes only a few seconds and thus does not affect scalability. Therefore, these two designs do not limit the scalability of our approach. We will discuss this more clearly in the final version.
>
> **Q4: Key‑term selection in Eq. 8 may be simplistic.**
>
> A4: Thank you for the question. Since the essence of retrieval is to match images and texts by computing the similarity between their cls token features, considering the attention weights from other tokens to the cls token alone can effectively reflect the importance of different tokens during keyword selection. When additionally considering the memory stored in the Key layer of the feed-forward network (FFN) for identifying input patterns [1], the performance improvement is limited. Specifically, according to the guidance in [1] for knowledge retrieval, the key most relevant to the cls token can be regarded as the one that best identifies its pattern. We compute the relevance between other tokens and this key, and use this value together with the attention weights to guide keyword selection. However, the experimental performance did not improve as expected and instead showed a slight decline, as shown in Table 3. We will provide a more detailed discussion of this in the final version.
>
> [1] Mor Geva, Roei Schuster, Jonathan Berant, and Omer Levy. "Transformer feed-forward layers are key-value memories." EMNLP 2021.
>
> **Table 3: Performance of CiPCL with additional factor**
> |Method|R@Avg|Iteration|Runtime|
> |:---|:---:|:---:|:---:|
> |CiPCL with Key memory|75.00|3521|09m56s|
> |CiPCL|75.15|3519|09m55s|
> ||||

---

> > ### Comment · Reviewer_vHM2 · 2025-08-03
> >
> > Thank you for addressing the main concern. Bumping the score to 5.

---

> > > ### Author Response · Authors · 2025-08-04
> > >
> > > Thanks for the recognition about our work.

---

### Decision · Program_Chairs · 2025-09-17

**Decision:**

Accept (poster)

**Comment:**

This paper analyzed several factors impacting cross-modal retrieval and found that  only token overlap degree correlates well to drop in performance. The authors then proposed two strategies to improve training efficiency:
1) a weighted sampling based on the token overlap degree, and
2) focusing on top-k critical word key in the source languages and only translating those into target languages.

Experiments on 2 benchmarks show that the proposed approach reduces the number of iterations by 30%-50% while maintaining performance. Reviewers recognize that this paper is clearly written with well-described problem statements, methodology, and contributions. However, there is moderate training speed gain and no quality gain.